# Identifying Birth Weight Cutoffs Based on Maternal Height and Apgar scores

Sumaiya Sultana Dola
*Applied Computer Science*
*The Univeristy of Winnipeg*
Winnipeg, Canada
dola-s@webmail.uwinnipeg.ca

Mir Md Taosif Nur
*Applied Computer Science*
*The Univeristy of Winnipeg*
Winnipeg, Canada
nur-m@webmail.uwinnipeg.ca

Camilo E. Valderrama
*Applied Computer Science*
*The Univeristy of Winnipeg*
Winnipeg, Canada
c.valderrama@uwinnipeg.ca

*Abstract*—The birth weight cutoff suggested by the World Health Organization (under 2500g) fails to reflect health risk across populations of diverse ethnicities. Based on that, previous studies have suggested using other indicators, such as maternal height and infant sex, to derive more accurate low birth weight (LBW) cutoffs. However, such approaches have not considered fetal well-being when deriving the cutoffs. Therefore, this study addresses this limitation with a novel approach using Apgar scores to derive LBW cutoffs based on maternal height. We used the 2022 CDC birth dataset to implement a two-stage analytical approach. The first stage used a Conditional Inference Tree (CIT) and a Fuzzy Inference Model (FIM) to identify combinations of maternal height and birth weight values associated with low Apgar scores. The second stage employed an ensemble of five machine learning regressors to estimate birth weight thresholds associated with normal Apgar scores. Our experimental results indicate that adaptive cutoffs outperform the fixed 2500-gram WHO cutoff. Specifically, the WHO cutoff does not effectively scale; its ability to detect newborns with low Apgar scores diminishes as maternal height increases. Overall, this research contributes to perinatal assessment by offering a method for identifying at-risk newborns based on maternal height and infant sex.

*Index Terms*—Apgar Score, Low Birth Weight, Fuzzy Inference Model, Conditional Inference Tree, Machine Learning Models.

## I. INTRODUCTION

Birth weight serves as an important indicator for assessing fetal well-being. It has long been utilized to identify at-risk newborns according to the guidelines set by the World Health Organization (WHO), which defines infants weighing under 2,500 grams as at risk [1]. While this threshold is widely accepted, it has faced criticism for its lack of adaptability, as it does not account for biological and demographic differences among various populations. This can lead to the misclassification of neonatal risks [2].

One critique of the WHO cutoff is that it was established based on cohorts of individuals of European descent, which limits its applicability to infants from diverse backgrounds [3]. As a result, previous studies have suggested the use of ethnicity-specific cutoffs. However, developing these specific cutoffs poses challenges due to globalization and the increasing number of mixed-race families [4]. In cases where parents come from different ethnic groups, determining the appropriate weight threshold can be problematic.

Given the limitations of ethnicity in determining low birth-weight (LBW) cutoffs, it is valuable to consider alternative factors that can provide more precise measurements. One such factor is maternal height, which has been shown to influence birth weight. Additionally, infant sex is another important consideration for defining LBW cutoffs, as previous studies has demonstrated significant differences in weight and other biometric measures between male and female fetuses [5]. On average, male newborns tend to have higher birth weights than females. However, despite their larger size, male infants consistently face a greater risk of mortality [6].

Using maternal height and infant sex, previous studies have defined LBW cutoffs [4], [7], [8]. Although these studies have analyzed the interaction between these two variables to develop percentile-based LBW cutoffs, they did not consider newborn health outcomes. This omission leaves uncertainty about whether the resulting cutoffs accurately reflect non-risk conditions. This limitation was highlighted in our previous study [9], where we reviewed births in the US in 2022 and found that newborns weighing less than 2,500 grams had a higher risk of adverse health outcomes when born to mothers taller than 163 cm, compared to those born to shorter mothers. This finding suggests that a fixed weight threshold does not adequately represent actual neonatal risk. This evidence highlights the importance of integrating newborn health status into cutoff estimations to define weight thresholds for each maternal height and infant sex that are associated with non-risk outcomes.

Our current study aims to fill this gap by combining Apgar score, an established indicator of newborn well-being, with maternal height and infant sex to determine low birth weight (LBW) cutoffs. To achieve this, we used a two-stage, data-driven approach that utilizes machine learning algorithms, moving beyond traditional statistical models. Our objectives are twofold: first, to investigate the relationship between maternal height, birth weight, and Apgar scores; and second, to establish birth weight cutoffs for maternal height that are associated with a normal Apgar score (NAS).

In summary, this study makes the following key contributions to perinatal care: (i) it provides empirical evidence advocating for a revision of the conventional LBW threshold; (ii) it demonstrates that, for a fixed birth weight, the likelihood

of achieving a normal Apgar score decreases with increasing maternal height; and (iii) it establishes maternal height– and infant sex–specific LBW cutoff values that are associated with normal Apgar scores.

## II. METHODS

### A. Dataset

For this study, we utilized the 2022 National Natality Dataset from the CDC [10], [11], which documents data on 3,676,029 births in the US in 2022. Each record includes 227 variables describing maternal, paternal, and neonatal characteristics, such as maternal height, weight, and ethnicity, along with birth outcomes including Apgar score, gestational age, and birth weight. These data were compiled from maternal admission forms and corresponding medical records obtained at the time of delivery.

From the available variables, we considered four variables: maternal height, birth weight, newborn sex, and Apgar score. We included only live singletons delivered exactly at 37 weeks of gestation, following the WHO criteria that states that newborns weighing less than 2.5 Kg at 37 weeks are at elevated risk [12]. Additionally, we restricted the analysis to 37 weeks of gestation to minimize the confounding effects, as birth weight naturally increases with advancing gestation. Twins and triplets were excluded due to their distinct growth patterns, lower gestational ages, and LBWs compared to singletons [13]. Records with missing data for any of the four selected variables were also excluded.

Maternal height was stratified into five categories based on the dataset distribution: 18.5% mothers were in <155 cm, 13.3% in 155–160 cm, 27.5% in 160–165 cm, 22.0% in 165–170 cm, and 18.8% in ≥170 cm. This stratification was designed as it reflected biologically significant thresholds, like short stature (<155 cm) and tall stature (≥170 cm). Moreover, we used 5 cm intervals (e.g., 165–170 cm) to ensure the height intervals were narrow, to ensure more accurate and closely related results within each group.

We used the 5-minute Apgar score as the outcome measure because it is a clinically validated indicator of immediate health condition. It directly reflects early neonatal risk related to birth weight [14]–[16]. We grouped them into two groups: "Low Apgar" (LAS) (<6 ), indicating serious health concerns [17], and "Normal Apgar" (NAS) (≥ 6), reflecting generally healthy conditions. This binary classification avoided sparse data in the moderate group, ensuring a consistent baseline for estimating minimum birth weight across maternal height and sex groups. Newborn sex was nearly balanced between females (49.9%) and males (50.1%) for the LAS group, whereas the NAS group had a higher proportion of females (58.1%) compared to males (41.9%).

## III. METHODOLOGY

Figure 1 shows the two-stage analytical approach used to examine the relationship between maternal height, neonatal birth weight, and Apgar score to determine appropriate birth weight cutoffs. Analyses were conducted separately for male and female newborns, given the known influence of sex on birth weight [8]. The following sections detail each step.

### A. Stage 1: Descriptive analysis

Stage 1 focused on performing a descriptive analysis using a conditional inference tree (CIT) and a fuzzy inference model (FIM) on 59,658 newborns with birth weights ranging between 800 and 2600 g. The CIT model generated a decision tree visualization, showing how different combinations of maternal height and birth weight categories were associated with Apgar scores. The FIM provided human-readable IF–THEN rules that summarized which combinations of maternal height and birth weight were most indicative of neonatal risk categories (Low, Medium, High).

*1) Conditional inference tree (CIT) :* The primary objective of implementing a CIT [18] is to partition a population into subgroups that show consistent outcomes regarding the clinical outcome of interest [19]. In this study, we trained CIT to classify the newborn APGAR scores, with its structure generated through recursive multiway partitioning and splits guided by statistical significance tests.

*2) Fuzzy inference model (FIM):* We used fuzzy logic in this study to generate rules relating birth weight and maternal height to Apgar score outcomes. Fuzzy logic allowed us to model these uncertain relationships more effectively by using membership functions that reflect various degrees of truth rather than relying solely on crisp variables [20].

*3) Training the CIT and FIM models:* To train and evaluate the CIT and FIM models, we split the 59,658 instances of the Stage 1 into training and testing sets. The dataset was divided into training (80%) and testing (20%) sets, with the test set reserved for performance evaluation on unseen data.

To optimize hyperparameters for both FIM and CIT models, we further split the training set into training and validation subsets. The configuration that achieved the best validation performance was selected for the final evaluation on the independent test set.

We marked LAS as "1" and NAS as "0" to assess the predictive models. The dataset was imbalanced, with approximately 3 percent of LAS cases occurring in males and about 2 percent in females. To balance the dataset, we used minority random oversampling.

As LAS cases represent a critical health condition requiring precise identification, we trained the CIT and FIM models with a priority on maximizing the prediction of LAS cases. As a result, the models prioritized true positive predictions, which led to a higher occurrence of false positives relative to false negatives.

*4) Performance evaluation:* We computed eight key metrics: specificity, sensitivity, geometric mean (G-Mean), receiver operating characteristic area under the curve (ROC AUC), area under the precision-recall curve (PR AUC), F1 score (weighted and macro), and accuracy to evaluate CIT and FIM. We calculated the G-Mean to ensure a balanced evaluation across both classes. Along with the macro F1 score, it helped reduce the bias caused by the larger NAS class, giving

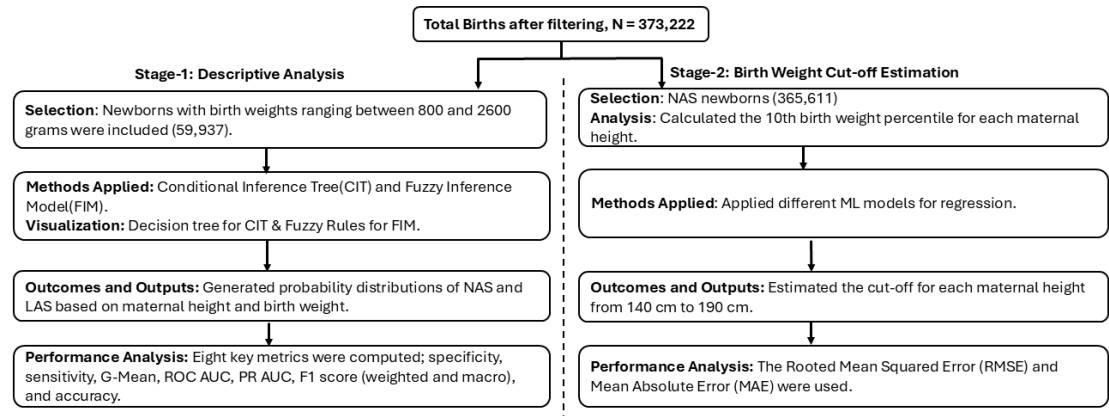

Fig. 1: Flowchart of the two-stage analytical method for estimating the cutoffs.

TABLE I: Evaluation metrics for FIM and CIT models for newborns risk classification (LAS and NAS cases)

| Model | Sex | Specificity (NAS) | Sensitivity (LAS) | G-Mean | ROC AUC | Accuracy | Weighted F1-Score | Macro F1-Score | PR AUC |
|---|---|---|---|---|---|---|---|---|---|
| **Female** | FIM | 50.8 | 65.4 | 57.7 | 61.6 | 51.2 | 65.7 | 36.4 | 4.6 |
| | CIT | 51.3 | 61.5 | 56.2 | 57.8 | 51.5 | 66.0 | 36.4 | 3.0 |
| | **Average** | **51.1** | **63.5** | **57.0** | **59.7** | **51.4** | **65.9** | **36.4** | **3.8** |
| **Male** | FIM | 60.9 | 54.8 | 57.8 | 58.1 | 60.7 | 72.9 | 41.6 | 5.1 |
| | CIT | 47.1 | 62.1 | 54.1 | 55.7 | 61.7 | 73.7 | 41.5 | 4.1 |
| | **Average** | **54.0** | **58.5** | **56.0** | **56.9** | **61.2** | **73.3** | **41.6** | **4.6** |

a fairer measure of how well the models performed across both NAS and LAS cases.

### B. Stage 2: Finding the birth weight cutoffs

Stage 2 focused on estimating birth weight cutoffs for NAS newborns. We used data from 365,611 NAS singleton infants born in the 37th week and calculated the 10th percentile birth weight for each maternal height. Then, we regressed the maternal height with the identified 10th percentile using multiple machine learning regressors, including support vector regression (SVR), extreme gradient boosting (XGB) regressor, random forest regressor (RF), ridge regressor, and fully connected neural networks (FCN). These models were applied to estimate flexible cutoff values for maternal heights between 140 cm and 190 cm.

*1) Regression models:* To capture both linear and non-linear relationships, we employed five regression techniques. These techniques included a Ridge Regression to model linear correlations while addressing potential multicollinearity by applying an L2 penalty. SVR with a Radial Basis Function (RBF) kernel transformed data into a higher-dimensional feature space, allowing non-linear patterns to be modeled. To enhance the predictive models' performance, we implemented ensemble approaches: RF and XGBoost, which used bagging and boosting algorithms, respectively. Finally, we applied an FCN to leverage deep learning to capture complex correlations between the 10th percentile birth weight and maternal height. To check for potential overfitting, we evaluated all five models using 5-fold cross-validation. In each fold, the input features and the 10th-percentile birth-weight cutoffs were standardized using only the training data, after which the model was fitted and its performance was assessed on the held-out fold. After

checking performance, we retrained the models on the full dataset to leverage all data for final estimations.

*2) Performance Evaluation:* For evaluating the regression machine learning models, we used Root Mean Squared Error (RMSE) and Mean Absolute Error (MAE) on the fitted lines.

*3) Birth weight cutoff estimation:* To determine the final birth weight cutoff for each maternal height, we aggregated the outputs of all five models. This ensemble approach captured the overall trend while minimizing dependence on the assumptions of any single model. Then, to capture the trends, the averaged estimations were fitted using a polynomial regression, modeling the estimated 10th percentile birth weights as a function of maternal heights for both male and female newborns. The degree of the polynomial was selected by minimizing the MSE. The polynomial coefficients were derived using the least squares method.

### IV. RESULT

#### A. Stage 1: Descriptive analysis

*1) Performance evaluation of FIM and CIT models on testing data:* Table I depicts the performance of the models on the independent test data. Both FIM and CIT showed a clear trade-off between sensitivity and specificity, mainly because of the class imbalance and the focus on prioritizing LAS detection. FIM achieved higher sensitivity for females (65.4%) but lower specificity (50.8%), while in males, sensitivity decreased (54.8%) with improved specificity (60.9%). CIT showed slightly higher sensitivity for males (62.1%) than for females (61.5%). FIM also demonstrated a marginally higher G-Mean compared to CIT. ROC AUC scores indicated moderate discriminative ability, whereas low PR AUC scores

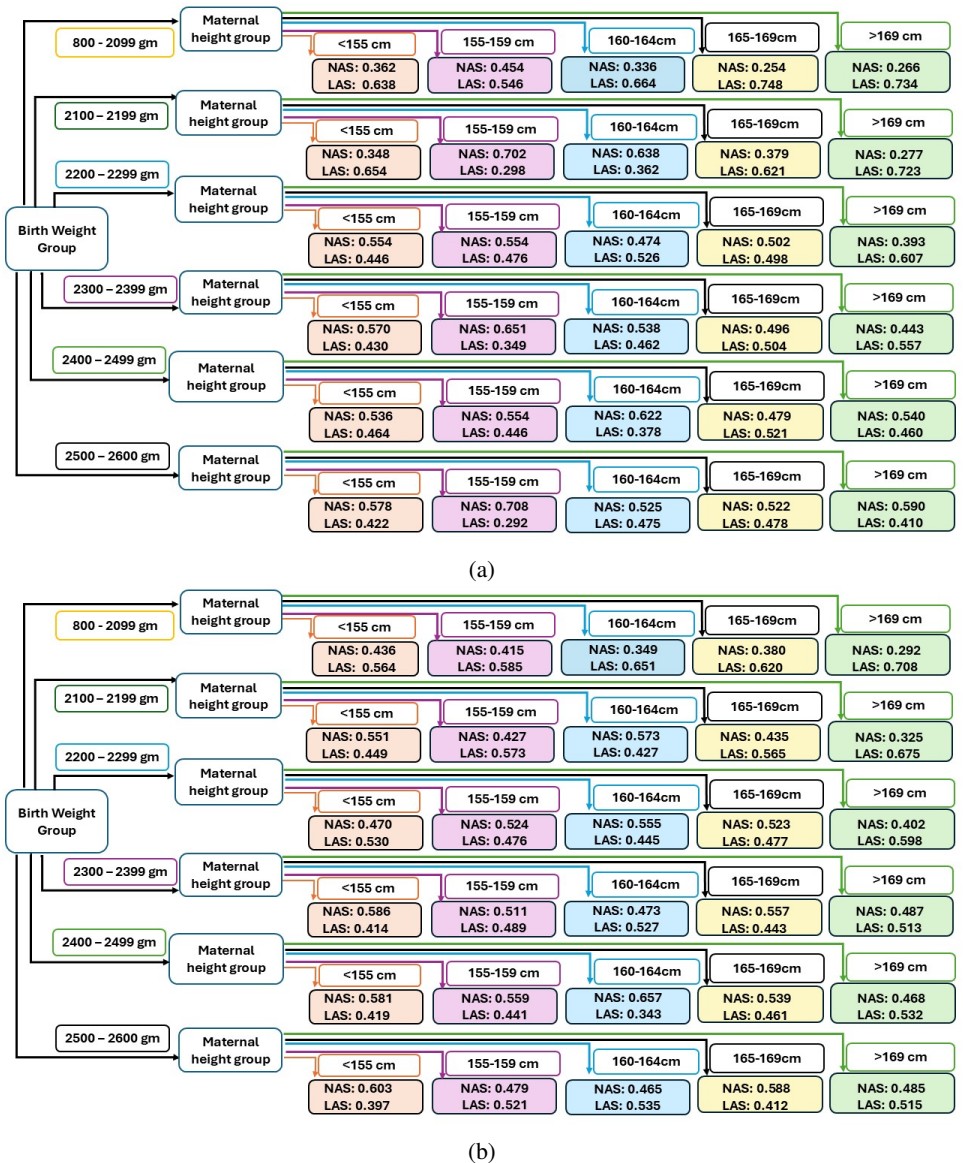

Fig. 2: CIT for detecting LA and NA scores among (a) male and (b) female newborns, based on the combination of maternal height (columns) and birth weight (rows).

reflected the challenge of maintaining precision and recall in an imbalanced dataset. The weighted F1 score (66%–74%) emphasized performance on the majority NAS class, while the Macro F1 score (36%–42%) provided a more balanced performance between LAS and NAS cases.

*2) Analysis of conditional inference trees for Apgar score prediction:* The analysis of CIT (Figure 2) to predict Apgar scores between male and female newborns shows how different combinations of maternal height and newborn weight influence the probability that a newborn will have LAS or NAS. The notable observation is that, for each birth weight group, as maternal height increases, the prevalence of NAS consistently decreases. Also, it is seen that, within any given maternal height range, as the birth weight of newborns rises, the probability of having NAS also increases. For instance,

among the maternal height range '<165 cm - 169 cm', as the newborn birth weight increased, the probability of having NAS also increased gradually.

*3) Fuzzy inference model for Apgar score prediction:* Table II lists the FIM rules for male and female newborns using three birth weight categories (Low:<2300 g, Medium: 2300–2500 g, and High: >2500 g), and three maternal height groups (Short: <155 cm, Medium: 155–165 cm, and Tall: >165 cm). The results show that when birth weight is low, the probability of a LAS newborn is higher for taller mothers and lower for shorter mothers.

### B. Stage 2: Finding the birth weight cutoffs

*1) Regression for estimating the 10th percentile birth weight associated with NAS newborns:* Table III shows the

TABLE II: Fuzzy IF–THEN rules and sex-specific LAS risk predictions

| Rule | IF Birth weight is | AND Maternal height is | THEN Male LAS risk | THEN Female LAS risk |
|------|--------------------|------------------------|--------------------|----------------------|
| 1 | Low ($<2300$ g) | Tall ($>165$ cm) | High | High |
| 2 | Low ($<2300$ g) | Medium (155–165 cm) | Medium | Medium |
| 3 | Low ($<2300$ g) | Short ($<155$ cm) | Low | Low |
| 4 | Medium (2300–2500 g) | Any height | Medium | Medium |
| 5 | High ($>2500$ g) | Any height | Low | Low |

summary of the RMSE and MAE of the five regressors used to estimate birth weight, showing performance on the full dataset, and on training and held-out test data using 5-fold cross-validation. For females, XGBoost achieved the lowest RMSE (10.33 g) and MAE (6.70 g). For males, XGBoost performed best with an RMSE of 11.14 and MAE of 7.24 g. When evaluated on the test set, both RMSE and MAE increased by approximately 8 points, indicating minimal overfitting. Although the gap was slightly larger for females, the difference was not substantial enough to suggest strong overfitting. Overall, the models appeared to capture meaningful patterns without simply memorizing the training data.

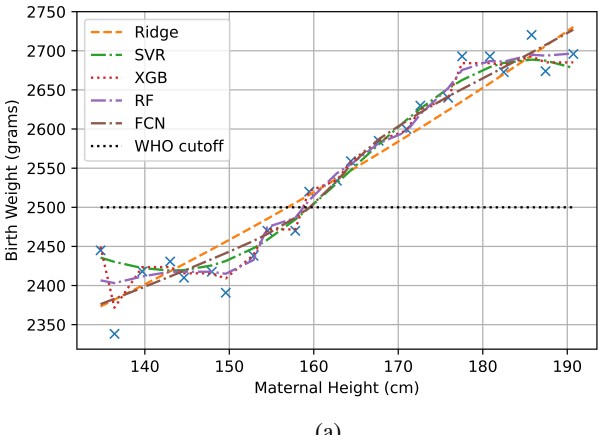

(a)

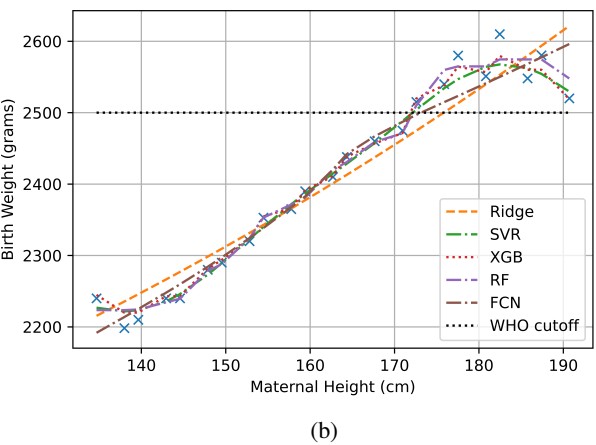

(b)

Fig. 3: Multiple regression analysis estimating the 10th percentile birth weight associated with NAS for (a) female and (b) male newborns across maternal heights.

Figure 3 shows the estimated 10th percentile birth weight for NAS newborns by maternal height. For male newborns with a NAS, the 10th percentile of birth weight remained below 2,500 g cutoff for mothers with a height range of 135 to 155 cm. At 135 cm height, it was around 2,400 g and was still corresponding to delivering a NAS newborn. As the maternal height increased to 160 cm, the estimated birth weight reached 2500 g. For mothers taller than 160, the 10th percentile birth weight for delivering a NAS newborn was higher than the WHO cutoff, ranging from 2500 to 2700 g. Among the regressors, XGBoost achieved the lowest RMSE and MAE.

In contrast, for females, the 10th percentile associated with NAS was below 2,500 g across a broader maternal height range, from 135 to 173 cm. At 135 cm, the threshold was approximately 2,250 g, notably lower than that of males. Beyond 173 cm, the required birth weight rose to between 2,500 and 2,600 g, still lower than the values observed for male newborns of taller mothers.

*2) Estimation of birth weight cutoff:* The degree 3 polynomial curves fitted on the predicted 10th percentile birth weight cutoffs across maternal heights based on the average outputs from the five machine learning regressors resulted in the following equations:

$$BW_{\male}(x) = -0.007x^3 + 3.40x^2 - 551.30x + 31847.62$$
$$BW_{\female}(x) = -0.005x^3 + 2.36x^2 - 374.55x + 21621.45, \quad (1)$$

where $x$ was maternal height in cm, and $BW_{\male}(x)$ and $BW_{\female}(x)$ were the estimated birth weight (in grams) for male and female newborns.

These equations estimate birth weight thresholds based on maternal height. For instance, a mother measuring 160 cm in height would need to deliver a female newborn weighing over 2,386 g and a male newborn weighing over 2,519 g to avoid a LAS. For both sexes, the equation indicate that as maternal height increases, the required birth weight for NAS also rises. Overall, for every additional centimeter of maternal height, the predicted birth weight cutoff increases by 5.5 grams for males and 7.2 grams for females.

*3) Comparing with WHO's fixed cutoff:* To assess the effectiveness of the proposed model using the WHO's fixed 2,500 g threshold, we compared both methods based on their ability to identify at-risk newborns with low Apgar scores. Figure 4 shows a recall comparison across maternal height and newborn sex. For both sexes, the WHO cutoff yielded higher recall within maternal height ranges corresponding to estimated birth weights below 2,500 g (approximately $<155$ cm for males and $<170$ cm for females). This higher recall

TABLE III: Summary of the RMSE and MAE of the five machine learning regressors used to estimate birth weight, showing performance on the full dataset, and on training and held-out test data using 5-fold cross-validation.

| Model | Male | | | | | | Female | | | | | |
|---|---|---|---|---|---|---|---|---|---|---|---|---|
| | Full Data | | Train (CV) | | Test (CV) | | Full Data | | Train (CV) | | Test (CV) | |
| | RMSE | MAE | RMSE | MAE | RMSE | MAE | RMSE | MAE | RMSE | MAE | RMSE | MAE |
| Ridge | 32.85 | 27.07 | 28.30 | 23.07 | 33.28 | 28.76 | 36.30 | 28.91 | 26.19 | 20.55 | 29.50 | 25.45 |
| SVR | 25.17 | 17.29 | 14.43 | 10.81 | 18.87 | 15.73 | 15.80 | 12.16 | 17.54 | 13.67 | 31.96 | 27.55 |
| RF | 19.71 | 13.86 | 15.10 | 11.38 | 22.77 | 18.15 | 14.83 | 11.16 | 19.29 | 14.42 | 30.75 | 27.53 |
| XGBoost | 11.14 | 7.24 | 13.09 | 9.40 | 30.23 | 26.57 | 10.33 | 6.70 | 16.76 | 11.52 | 37.17 | 33.72 |
| FCN | 27.40 | 20.94 | 21.16 | 15.90 | 29.59 | 25.41 | 27.27 | 18.80 | 18.50 | 13.05 | 22.21 | 17.56 |
| **Average** | **23.25** | **17.28** | **18.81** | **14.51** | **26.95** | **22.52** | **20.90** | **15.54** | **19.26** | **14.64** | **30.72** | **26.76** |

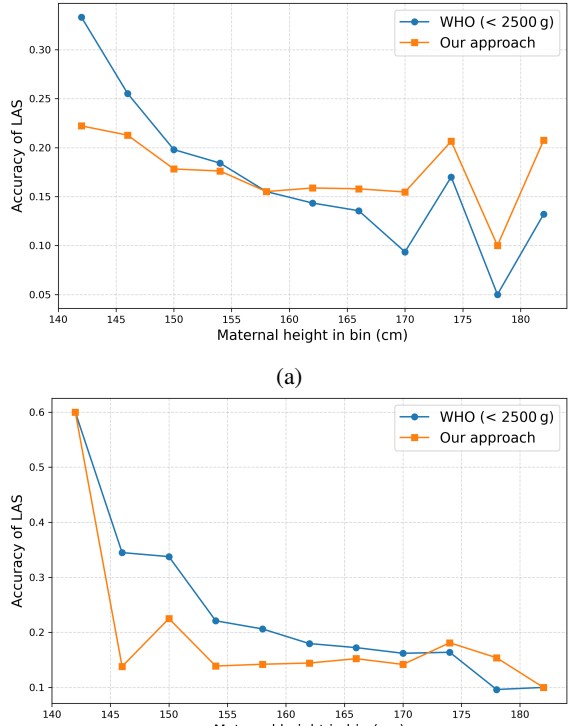

(a)

(b)

Fig. 4: Recall (TPR) of identifying at-risk newborns (Apgar <7) using WHO's fixed cutoff (<2500g) versus our maternal height-adjusted cutoff, stratified by maternal height for (a) male and (b) female newborns.

stemmed from the WHO criterion classifying a broader group of newborns as at-risk, thereby capturing more true positives, but also increasing false positives. However, for taller mothers, the WHO threshold often failed to detect newborns with low Apgar scores. In contrast, our adaptive cutoff provided more consistent recall across all maternal height ranges and both sexes, reducing false negatives and offering a more accurate reflection of actual neonatal outcomes.

## V. DISCUSSION

### A. Main Findings

The primary goal of this study was to develop a data-driven framework for estimating flexible birth weight cutoffs associated with a normal Apgar score based on maternal height. Our findings show that maternal height significantly influences cutoff estimation, with Stage 1 models revealing that newborns from shorter mothers often had NAS, even when weighing less than 2500 g, while newborns from taller mothers often needed higher birth weights to consistently obtain NAS. These results show that the fixed WHO's cutoff fails to detect at-risk newborns, and that adaptive thresholds based on maternal height offer a more accurate way to assess newborn risk.

Based on the evidence obtained from Stage 1, in Stage 2, we used multiple machine learning regression models to determine the birth weight thresholds using maternal height and Apgar score. The derived equations state required cutoffs for NAS newborns at 37 weeks increase with maternal height, often exceeding 2500 g, while thresholds for shorter mothers remain below this value. This highlights that the fixed 2500 g standard does not adequately reflect the risk profiles for mothers of different heights, as it underestimates risk for taller mothers and overestimates it for shorter ones. Furthermore, we found that female newborns require relatively less birth weight than male newborns to be considered NAS. This can be explained by the anatomical difference between sexes [5] and the higher mortality associated with males [6], [15], thus suggesting that a male newborn requires a higher birth weight to be considered a low-risk condition.

### B. Comparison with previous works

Previous research has highlighted the need for flexible birth weight thresholds based on ethnic factors [2], [3]; however ethnicity-specific studies are now less common due to the rise of mixed-race families. Our study focuses on maternal height as an anthropometric factor to evaluate neonatal health outcomes, addressing gaps in earlier studies that excluded newborn health status such as Apgar Score [4], [7], [9]. By incorporating Apgar scores, we created a more comprehensive and clinically relevant framework for deriving flexible LBW cutoffs based on maternal height and newborn sex, avoiding misclassification linked to the fixed 2500g threshold.

Additionally, to the best of our knowledge, this is the first study to use a data-driven approach to derive LBW cutoffs associated with NAS. We combined interpretable ML models (CIT, FIM) to visualize relationships between maternal height, birth weight, and neonatal health, and applied multiple regression models to estimate LBW cutoffs. This approach improves

clarity and clinical applicability, addressing limitations of previous static analyses that overlooked neonatal health status.

### C. Limitations and Future Work

We note that our current work faces some limitations. First, our analysis was restricted to data from the US population, which may limit how well the results generalize to other regions. However, because the US population is highly diverse, the approach we propose could be applicable to other countries. Future research should confirm these findings across different countries and healthcare systems. Second, while factors like maternal BMI, age, nutrition, and smoking could enhance prediction, we intentionally excluded them to isolate the effect of maternal height. This focus provides a simple, practical proxy for identifying potential LBW cases, particularly for resource-limited settings where detailed maternal data may be unavailable. Third, we could not assess paternal height due to data limitations; however, prior research suggests maternal height is a stronger predictor of birth weight than paternal height [21]. Finally, neonatal health was assessed using Apgar scores, which may not fully capture all morbidities or long-term outcomes. Future work should validate our result using more detailed measurements, such as cord blood gases or follow-up data for improved risk stratification.

The models showed relatively low accuracy, which was expected since the goal was to demonstrate that maternal height shifts the LBW cutoff compared to WHO's fixed threshold, rather than to build a high-performance classifier. Using only maternal height, birth weight, and Apgar score limits predictive power, while other factors (e.g., pregnancy complications, growth restrictions) further complicate accurate risk discrimination by influencing [9].

## VI. Conclusion

This work presents a two-stage computational framework that integrates explainable models and machine learning regressors to derive adaptive birth weight cutoffs based on maternal height and newborn sex. Our results show the limitations of using a fixed LBW threshold, which overlooks maternal anthropometric variation and can misclassify neonatal risk. In contrast, adaptive, data-driven cutoffs generated through our pipeline provide a more accurate and fair approach for assessing newborn health. These findings emphasize personalized, informatics-driven health standards that enhance the accuracy of neonatal risk assessments and support future enhanced healthcare applications.

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
