# OpenReview forum: "Identifying Birth Weight Cutoffs Based on Maternal Height and Apgar scores"
_IEEE.org/EMBS/BHI/2025/Conference — BHI 2025_

### Official Review · Reviewer_iU4X · 2025-07-13
**Using such population is goog and the paper is well structured.**

**Confidence:** 4
**Clarity Of Writing:** good
**Clinical Significance:** good
**Methodological Novelty:** good
**Overall Rating:** 7
**Final Rating:** 7

**Experiments And Results:**

good

**Questions For The Authors:**

- How can clinicians use the outcomes of your research? The paper could provide a concise discussion on how these findings could be applied in real-world clinical settings.
- While using the presented dataset is interesting, how would you expect the methodology to perform on different datasets, even within the US? How robust is the model to variations in other datasets? It might be useful to discuss this issue to strengthen the generalizability of the model.

**Strengths:**

- The paper is well structured.
- It addresses an important topic: how these cutoffs should vary rather than being a single number across different ethnic groups.
- The sample population is reasonably acceptable for this kind of outcome research.

**Summary Of The Paper:**

Using a data-driven strategy, this paper investigates new approaches beyond traditional models for determining Low Birth Weight (LBW) cutoffs. The estimation is based on maternal height and infant sex. The relationship among maternal height, birth weight, and Apgar scores is evaluated. Different machine learning classifiers are employed.

**Weaknesses:**

- Discussion Section B, Comparison with Previous Works: The paper compares its outcomes with previous studies, but no specific references are cited in this section for readers to consult. The relevant prior works should be properly cited here.
- Relying solely on the Apgar score should be acknowledged as a limitation, since other clinical measurements could also serve as strong predictors of newborn health.

---

### Official Review · Reviewer_ma9u · 2025-07-15
**New cutoff for neonatal weight to predict health of newborns**

**Confidence:** 4
**Clarity Of Writing:** great
**Clinical Significance:** great
**Methodological Novelty:** excellent
**Overall Rating:** 6

**Experiments And Results:**

great

**Questions For The Authors:**

What other methods did you consider to infer the probability distribution for NAS and LAS groups. Why did you settle on a combination of CIT and FIM?

**Strengths:**

- I think a good justification is provided to support the premise of this paper
-The methodology is robust
- The paper identifies limitations and potential enhancements to its premise

**Summary Of The Paper:**

This study challenges the sufficiency of the universal birth weight standard, positing that it overlooks significant determinants such as maternal height. Through a two-stage analytical framework that leverages machine learning models and the 2022 CDC birth dataset, the research investigates the intricate connections between maternal stature, neonatal birth weight, and Apgar scores. The findings advocate for personalized birth weight evaluations, which incorporate both maternal height and infant sex, as a more precise method for assessing neonatal health risks and ultimately fostering better health outcomes.

**Weaknesses:**

- I don't understand the justification for the 10th percentile cutoff
- The abstract of this paper mentions race as an important consideration, but ignores it for the rest of the paper. I think the story of the paper would flow better with a focus on maternal height and why that is an especially important consideration. Justify why only the maternal height was considered.
- Can you add more stats about how much this model outperforms the WHO standard? This piece is crucial to convincing the reader of your story

---

### Official Review · Reviewer_xvST · 2025-07-17
**Personalised identification of normal birth weight cut-off estimation**

**Confidence:** 3
**Clarity Of Writing:** great
**Clinical Significance:** great
**Methodological Novelty:** good
**Overall Rating:** 5
**Final Rating:** 6

**Experiments And Results:**

fair

**Questions For The Authors:**

How balanced was the dataset when it came to maternal heights?

The authors state the percentage of male and female newborn babies in LAS cases. On the other hand, what  is the gender distribution for NAS cases or overall?

The model showed higher sensitivity and accuracy for male infants. While not necessarily, this could suggest a sex-based imbalance or bias towards male babies. How do you comment on this finding?

The authors excluded twins and triplets from their analysis, could the reasoning behind that decision be explained?

Do the authors plan on applying their results in a clinical setting? And if so, how would they go about it?

**Strengths:**

Strong clinical impact: strong potential to improve neonatal risk assessment.
Incorporation of Adgar scores: the inclusion of an indicator of newborn’s health increases the clinical impact of this study.
Sex-specific analysis: consideration of biological differences is a plus accounting for any gender biases
Robust approach and clear methodology structure: all analytical tools and models used in the scope of this study were appropriate in regards to their aim and they mostly succeeded in clearly communicating their methods and limitations.

**Summary Of The Paper:**

The paper challenges the idea that all newborns under 2500 grams should automatically be considered at low-weight and at risk, as defined by the WHO. They explored how factors like maternal height and Apgar score might influence this fixed threshold and they aimed to develop a more personalized and adjustable low birth weight (LBW) cutoff. They applied a two-stage method using the CDC dataset. The first stage was descriptive analysis to explore the relationship between maternal height, birth weight, and Adgar score, using the CIT model and fuzzy rules logic and categorizing NAS and LAS cases. Then, they applied 5 regression ML models to find the sex specific LBW cut-offs for various maternal height ranges. Finally, degree 3 polynomial equations were fitted to the ensemble predictions to obtain a continuous and sex-specific function to calculate the LBW cutoff estimations across maternal heights. Their results show that to determine if a baby is underweight or not, more factors should be considered. For example, maternal height as shorter mothers might give birth to healthy babies that are under the current LBW limit set by the WHO, while taller mothers might give birth to babies that need to exceed the 2500 g limit to be considered as healthy. In short, they propose that to better reflect the baby’s health, a more personalised cut-off value for normal birth weight should be taken into consideration.

**Weaknesses:**

Generally, while there were some limitations like imbalance between NAS and LAS cases, limited generalizability (only U.S. data), etc., the authors were transparent about the limitations of their study. However, I do have some concerns:

While the models are useful, they may be too complex for easy use in clinical environments without a supportive digital tool.

Accuracy is notably low and there is no clear explanation as to why performance is so low.

---

### Official Review · Reviewer_3d3g · 2025-07-17
**Review for "dentifying Birth Weight Cutoffs Based on Maternal Height and Apgar scores"**

**Confidence:** 3
**Clarity Of Writing:** great
**Clinical Significance:** good
**Methodological Novelty:** great
**Overall Rating:** 5

**Experiments And Results:**

good

**Questions For The Authors:**

1. Have you examined whether your height-based LBW cutoff equations hold in populations outside the US, or using data from other years or regions? If not, what are your thoughts or plans regarding validating these models in other demographic or socioeconomic contexts?
2. Can you explain why you restricted your analysis to exactly 37 weeks' gestation, and do you believe your approach would generalize to other gestational ages (e.g., preterm or later-term births)? If you have conducted any preliminary analyses or have plans to do so for other gestational groups, please elaborate.
3. Did you assess the influence of factors such as maternal BMI, parity, comorbidities, or socioeconomic status? If constrained by data limitations, can you discuss potential confounders and their likely effect on the height-birthweight-Apgar relationship? Are you planning to test the stability of your model with these variables when data become available?

**Strengths:**

1. Innovative and Patient-Centered Methodology
The study moves beyond traditional, universal, or ethnicity-based birth weight cutoffs and proposes a customized LBW threshold based on maternal height and newborn sex, which is more reflective of individual risk and diversity.
2. Integration of Health Outcome (Apgar Score)
The methodology anchors the new cutoffs to actual newborn health status (Apgar score), rather than relying purely on population-based percentiles. This ensures the proposed thresholds are clinically meaningful and relevant to real neonatal risk.
3. Large, High-Quality Dataset
The analysis uses the 2022 CDC National Natality dataset, comprising millions of births, which ensures high statistical power and generalizability within the US context. Robust data selection and cleaning (restricting to singletons at 37 weeks with complete data) strengthen the analysis.
4. Robust, Modern Analytical Approach
The paper leverages machine learning models and ensemble techniques (including both interpretable and predictive models) to explore complex relationships and validate findings across multiple algorithms, improving confidence in the results.

**Summary Of The Paper:**

Low birth weight (LBW, <2,500g) is widely recognized as a risk factor for neonatal morbidity and mortality, but the World Health Organization’s global standard does not account for geographic, ethnic, or biological diversity. With increasing population admixture, ethnicity-based standards are less useful. Maternal height is a known determinant of birth weight, but current cutoff models do not consider neonatal health (as measured by Apgar scores). The objective of this paper is to develop birth weight cutoffs for LBW that are tailored to maternal height and infant sex and validated against neonatal health status (Apgar score), rather than relying on universal or ethnicity-based standards. The study demonstrates the inadequacy of a single, universal LBW cutoff. Adjusting LBW thresholds for maternal height and infant sex—validated by neonatal health status—results in a more accurate and clinically meaningful assessment of at-risk newborns. The provided polynomial equations allow for individualized evaluation and have potential to improve perinatal care and outcomes.

**Weaknesses:**

1. Limited Generalizability Due to Single-Country, Single-Year Dataset
The analysis is limited to the 2022 CDC US natality dataset, with no external validation on other populations. Thus, it is unclear whether the findings apply to other countries, regions, or healthcare systems, particularly populations with different socioeconomic, nutritional, or health characteristics. The authors can try to replicate their findings using natality data from other countries or demographic groups (e.g., Europe, Asia, lower-resource settings). Such external validation would demonstrate the generalizability and robustness of the proposed height-based cutoffs, and may reveal population-specific differences or identify necessary adjustments for broader use.

2.The study restricts analysis to singleton, term infants born specifically at 37 weeks. This limits the usefulness in clinical practice, where gestational age varies and the relationship between maternal height, birth weight, and outcomes may shift at different gestations (preterm or later-term babies). The authors can consider extend the approach to include other gestational ages (e.g., 38–41 weeks, preterm births), performing similar modeling to derive gestational-age-specific equations. This would enhance the clinical utility and relevance of the cutoff equations to a much wider clinical population, and clarify whether the maternal height-birthweight-Apgar relationship is consistent or varies by gestational age.

3. The models consider only maternal height, birth weight, newborn sex, and Apgar score. Other well-established determinants of birth weight (maternal BMI, nutrition, socioeconomic factors, smoking, parity, comorbidities, and even paternal height) are not included, possibly confounding the results.

Minor Comments:
1. The study dichotomizes the 5-minute Apgar score at <6 versus ≥6, treating ≥6 as "normal." However, more granular stratification is common in neonatology (e.g., <4 severe, 4–6 moderate, 7–10 normal), and Apgar does not capture all possible forms of neonatal morbidity. Additionally, Apgar does not address longer-term health outcomes.
2. While ensemble machine learning models are used to capture complex patterns, the manuscript underreports model interpretability and does not discuss possible overfitting, despite the large dataset. Some models (e.g., neural networks) are much less interpretable for clinicians. Maybe include SHAP (SHapley Additive exPlanations), partial dependence plots, or similar techniques to clarify the relative importance of variables and patterns learned by black-box models. Discuss overfitting by showing how models perform on held-out/external validation data. This would improve clinician trust, facilitate adoption, and help researchers understand underlying mechanisms.